# Towards a Multi-Scale Effect of Land Mixed Use on Resident Population—A Novel Explanatory Framework of Interactive Spatial Factors

**Liu Liu [1,2], Huang Huang [1,3,*] and Jiaxin Qi [1]**

1    Department of Urban Planning, College of Architecture and Urban Planning, Tongji University, Shanghai 200092, China; l.liu@tongji.edu.cn (L.L.); jiaxin0328@tongji.edu.cn (J.Q.)
2    Key Laboratory of Spatial Intelligent Planning Technology, Ministry of Natural Resources, Tongji University, Shanghai 200092, China
3    Key Laboratory of Ecology and Energy Saving Study of Dense Habitat, Ministry of Education, Tongji University, Shanghai 200092, China
*    Correspondence: hhuang@tongji.edu.cn

**Abstract:** Starting from Jane Jacobs' critiques and largely promoted and emphasized by New Urbanism, land mixed use (LMU) has become prevalent worldwide. It is believed to be an efficient approach to shaping a higher level of vitality in the economy, equality in society, and quality in the environment. To reveal the differences of this effect at distinct spatial scales, this study selected the two most related outcomes of LMU—resident population distribution and changes—to investigate the LMU impacts. A novel framework is developed to quantify the interactive impact of pairwise LMU-related factors at multiple scales, and the geographical detector is applied to identify the relationship between resident population distribution/changes and LMU. Taking the Jiading District of Shanghai as a pilot case, the framework was applied and tested. The results showed LMU affected resident population distribution distinctively from 600 m to 3000 m grid scales. The grid scale of 1800 m, approximately ten blocks, is revealed to be the optimal scale for discussing LMU with the selected factors. Also, these factors play different roles at different spatial scales. Some factors strongly affect the resident population distribution only when working with other factors. The study emphasized the crucial role of scale in LMU and suggested an open framework to support the decision making and policy making in planning for a better performance of smart growth and sustainability via LMU. It can help researchers obtain the optimal scale for the LMU plans with different sets of factors and identify the key factors in various contexts. Thus, this framework also contributes to supporting other practices of land mixed use beyond our study region.

**Keywords:** land mixed use; multiple scales; pairwise factors; geographical detector analysis; resident population distribution

## 1. Introduction

Land mixed uses (LMUs) have been a prevalent mantra to support the formation of a higher level of vitality in the economy, equality in society, and quality in the environment [1–3]. It has become a central principle to reach 'smart growth' and sustainability, promoted by New Urbanism [1,2,4–9]. In contrast to land-use separation that evolved into early zoning regulations and planning single-purpose districts [10] and thrived in the early twentieth century, the operational concept of LMU has yet to evolve [9]. Just as Raman and Roy suggested, although LMU has been widely accepted and largely promoted through almost all international forums and academic discourses, confusion and gaps in terms of operation remain.

The early discussion of LMU mainly focuses on the types and proportions of the land mixture as well as corresponding effects at the macroscale and compatible uses at the

neighborhood scale [1]. The mixed uses of business, recreation, institutions, and residence have been the most widely discussed functions [1,11]. Although industrial use has been less commonly included, the current study also suggests the possibility of a mixture in certain urban regions [11]. The degrees of the land mixture in terms of land-use priorities, distance, and attributes are taken as key factors in the existing research to reflect the outcomes and direct relevant practices [12,13]. More recently, the discussion also extended to travel behavior that is closely related to health outcomes [14], life satisfaction [15], and energy and environmental performance resulting from land mixed uses [16], which were highly related to timely topics and current development challenges.

As the existing studies suggested that simple, area-based measures of land use mix are not sufficient to capture the complex difference resulting from the nuance of land use mixture [14], indexing is, therefore, one of the approaches that are most commonly employed and suggested methods to analyze the effects of LMU [9,11,12,17]. In the same light, more complex mixed-use indicator sets were introduced for the studies at different spatial scales (e.g., city, district, subdistrict, or neighborhood) and in various classifications (e.g., land categories) [13]. Additionally, the development of techniques in terms of analyzing abilities and methods as well as obtaining more varied data allows the search for fine-grained patterns of LMU [11,13], multi-sourced research [13], and introducing new approaches such as genetic algorithms and models to quantifying mixed-use effects, among others [16].

Nonetheless, there are still noticeable gaps between theoretical claims and practical outcomes of LMU resulting from a lack of comprehensive understanding of types of LMU, patterns of mixing, the safety of mixing, and the applicability of mixing at various spatial scales [9]. LMU could also result in less ideal states in practice, including uneven development at both macro- and micro-scales for various reasons [18]. Thus, scale, density, and the degree of the mix were identified as the essential factors among the other numerous factors affecting the development of mixed land use [9,19].

Therefore, by focusing on 'scale' among the other two essential factors, this study proposes a novel explanatory framework to research the effect of LMU on resident population distribution and changes at multiple spatial scales. Rather than selecting different index frameworks or sets of indicators that are developed from various mixed-use aims and their corresponding analyzing methods as well as sources of appliable data, this study aims to reveal the effect of mixed-use by conveying the impact of a fixed set of land-related factors and to scrutinize what is the most applicable scale for revealing LMU impacts. Thus, the research hypotheses are set as follows:

A fixed set of LMU-related factors shows distinct effects of resident population distribution and changes on different spatial scales.

There is an optimal scale for discussing a fixed set of LMU factors.

Identifying the scale can provide more accurate support for spatial interventions of efficient LMU.

By employing resident population distribution and changes as an outcome of the land use mix, our analysis focused on the interconnections between functions (e.g., work, live, and visit) rather than the single function itself, as Dovey and Pafka suggested [20]. In addition, it also helps us to develop criteria for identifying and confirming the selected set of LMU factors to test our explanatory framework proposed in the following section.

## 2. Methodology Framework

### 2.1. Theories and Methods

To verify our hypothesis, we fully considered the Modifiable Areal Unit Problem (MAUP) related to different spatial scales, especially at the neighborhood level. An analysis tool named Geographical Detector (GD) was applied in this study to characterize the impact of distinct land-use factors on the resident population.

(1) Modifiable Areal Unit Problem.

As population can be aggregated from a basic unit to a given scale, MAUP needs to be considered for the analysis of such spatial correlations [21]. According to MAUP, the results

of spatial analysis are sensitive to spatial units and spatial division ways, i.e., scale effect and zoning effect [22–24]. The scale effect relates to different geographic units, such as the resolution of raster data or the administrative boundary of different levels (ibid). Thus, it is important to select a proper scale for spatial analysis, which is also an open question. In this paper, we selected an appropriate scale by comparing the influence of different factors at multiple scales.

(2)   Geographical detector.

Geographical detector (GD) is a spatial statistic method that explores the spatial heterogeneity of geographic phenomena and reveals their driving factors [25,26]. It is built on the assumption that the spatial distribution of a dependent variable and an independent variable should be similar if the independent variable has a significant impact on the dependent variable. The GD consists of four detectors, i.e., factor detector, risk detector, ecological detector, and interaction detector. Geographic detectors can detect possible causal relationships between two variables by examining the coupling of their spatial distributions.

A factor detector relies on a statistic q [25] that shows how factor X can interpret the spatial heterogeneity of a dependent variable Y. According to [25,26] the q-value is computed with Equation (1):

$$q = 1 - \frac{\sum_{h=1}^{L} N_h \sigma_h^2}{N\sigma^2} = 1 - \frac{SSW}{SST} \tag{1}$$

$$SSW = \sum_{h=1}^{L} N_h \sigma_h^2, \ \ SST = N\sigma^2$$

where h represents the so-called strata of X. In Equation (1), SSW means 'Within Sum of Squares' and SST is the 'Total Sum of Squares' [25]. The range of the q-value is between 0 and 1, and a larger value of q indicates X has higher explanatory power than Y. A statistical test of significance is applied to the q-value of each X.

An interaction detector can be used to evaluate the overall effect of two factors on Y, i.e., whether their combination increases or decreases their explanatory power [25]. It takes the strata of X1 and X2 and computes their intersection, i.e., new strata derived from X1 ∩ X2. Correspondingly, a q-value is calculated for X1 ∩ X2. According to the comparison of q(X1)/q(X2) and q(X1 ∩ X2), five types of effects are defined for interaction [25]. A complete explanation can be found in relevant references [25,27].

(3)   Problem Definition.

In this paper, we define the problem as how the resident population distribution and change of an area can be influenced by LMU on different spatial scales. Subject to the MAUP effect, the correlation and/or interactions between LMU factors and resident population distribution could vary across different spatial scales, and we defined it as a multi-scale effect in this paper.

We investigated the multi-scale effect on resident population distribution driven by different spatial factors and their interactions. Two dependent variables, Y1 and Y2, stand for the 2020 resident population distribution and its 10-year population change, respectively. We selected a set of 14 independent factors that are highly relevant to resident population distribution in the study area. They consist of two representative groups: the first refers to the amount of Road length, Bus stops, Shopping malls, Enterprises, Schools, Hospitals, or Parks and Squares in a grid, and the second group is about the minimum distances from a grid center to metro stations, bus stops, enterprises, hospitals, schools, or parks and squares. The first group represents the density of PoI in a spatial unit and reflects its primary functionality, while the second group reflects the accessibility of the spatial unit to different facilities. For instance, the minimum distance to metro stations of a grid can indicate its geographic closeness to access metro stations, which may considerably influence its attraction to residents.

Based on grid models of the study area, we mainly employed the factor detector and interaction detector of GD to reveal the significance of different independent variables (factors) and their pairwise interactive impact on the dependent variable (Y1 and Y2). The q statistic of GD reflects the explanatory power of individual or pairwise factors. We visualized and compared the q-values on different scales to depict the spatial correlations between Y1/Y2 and these factors, which supports the analysis of the multi-scale effect of land use mix on the resident population.

### 2.2. Framework

As mentioned above, we aimed to reveal the effect of land use mix on resident population distribution and changes at different scales. An open framework is proposed to verify the research hypothesis. The research consists of two phases (Figure 1). In phase 1, we conducted a conceptual analysis for the research object (resident population distribution/change in our case), including the collection and processing of multi-sourced spatial-temporal data and the selection of a representative factor set. Specifically, the distribution of the resident population at different years and their difference/change can be represented in a grid model. Attribute statistics can be conducted in this discrete spatial model to fuse different data. To consider LMU, the selected factors defined with corresponding PoI data are reflected in the grid model as well.

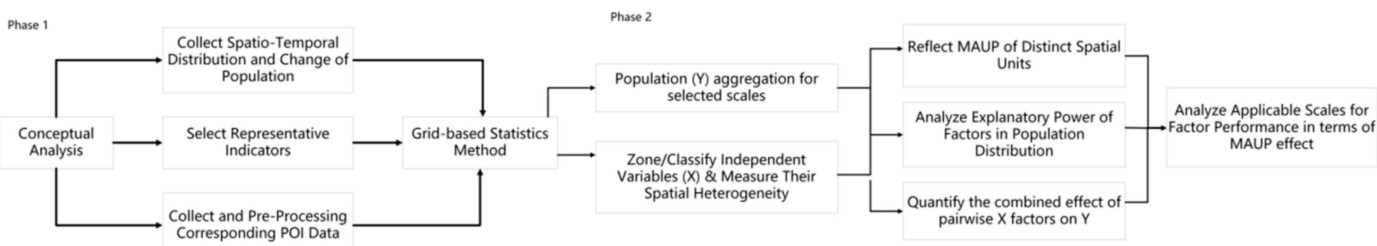

**Figure 1.** The proposed framework. Source: Authors' construction.

In phase 2, we aggregated the grid models with the investigated attribute (e.g., resident population) at given scales and zoned the models with the values of the attributes of the selected factors. This measure helps to better reflect the spatial heterogeneity of the factors. With the support of the GD, the explanatory power of single factors for Y1/Y2 can be calculated. Accordingly, we can analyze the MAUP effects within distinct spatial scales with the explanatory power of the factors, which also provides a rank of independent factors' influence. By comparing the contributions of different LMU factors, we can confirm the applicable spatial scales of all the selected factors and identify the optimal scale for them.

Furthermore, we can examine the combined explanatory power of pairwise factors on Y1 and Y2. This measure can explicitly assess the influence of the land use mix indicated by both factors. By checking the interaction between two factors, we focus on the non-linear enhancement of the factors and compare their independent influence. In this way, we can mine the potentially important factors for Y1/Y2, even though they are not the most influential factors.

This explanatory framework aims to provide an analysis solution for any specific research region, which can support urban planners in reviewing the optimal spatial scale for plans with the selected land use factors. It also helps the planners identify and confirm the key factors and their interaction effects on resident population distribution or another given context.

### 2.3. Data Requirements

(1)    Data Types.

Recently, there have been LMU-related studies that consider LMU influence and its correlation with other factors of human activities and built environments. By using PoI (point of interest) data and OpenStreetMap (OSM) street networks, researchers can identify

the functional zoning and the mixing degree of the central area of Chongqing City [28]. Some other researchers [29] employ different indicators of population density, floor area ratio, business POIs, road network, and built-up area to investigate the pattern of land use with population and economic activities and derive the spatial aggregation patterns of these indicators. Therefore, PoI data regarding public facilities are an important data source for depicting LMU.

As we plan to examine the relationships between resident population and LMU-related factors, the geospatial data of population and public facilities are employed. Three types of data are applied in this study. The first one is the open population data from WorldPop. As the worldpop dataset was generated in a 'top-down' estimation whose most important source is population and housing censuses, we interpret the data as resident population.

In addition, we adopted other open geospatial data, including PoIs from Alimap and the latest road networks from OSM. PoI mainly refers to the configuration of public service facilities, and LMU can be reflected by PoI combinations in spatial grids. In addition, OSM provided accurate urban road networks. For OSM road types, we excluded internal roads and sidewalks, such as those with the tags of 'path', 'service', 'footway', 'pedestrian', and 'steps'.

We selected a regular-grid model to fuse all the spatial data, which can facilitate the comparison between different spatial scales. As the research aims to investigate the multi-scale effect of LMU factors, the computation of GD relies on different rasterizations of a geographic area with distinct grid sizes.

(2) Define LMU factors.

As mentioned before, we defined LMU factors/indicators with PoI data. Considering availability, relative accuracy, relevance, compatibility, and other issues of data, we identified 14 factors for GD computation (see Section 3.2). Basically, they are separated into two groups. One is about the total number (N) of PoIs on one grid (i.e., density), and the other is about the minimum distance (MD) from the grid center to PoIs.

By exploring the multi-scale effect of LMU on resident population distribution, we also try to reveal the most applicable scales for certain patterns of mixed factors. Therefore, instead of identifying the 'best' set of influential factors, our focus is to compare the performance of a given set of factors at various spatial scales. In this way, we can obtain a better understanding of the specific contribution of mixed factors to population issues (i.e., population distribution Y1 and population changes Y2).

## 3. Pilot Case Study

### 3.1. Case of Jiading and Its Development Trajectory

Jiading District was selected for this pilot case study because it is listed in the earliest plan of Shanghai, and it evolved from a satellite city into a new district of Shanghai with a new town that functioned relatively independently (e.g., public services, housing, and/or job opportunities), though it is not yet a regional center (see Figure 2). Jiading, therefore, shows a comparatively clear trajectory of development from segregated land use towards an independent urban region with comprehensive functions (Figure 3). LMU is an important approach implemented during this transformation [30] (also see Figure 4). Additionally, attracting the resident population to settle in Jiading and balancing its distribution is one of the overarching aims, which also fits our research design that applies resident population distribution and changes as a reflection on the effect of LMU.

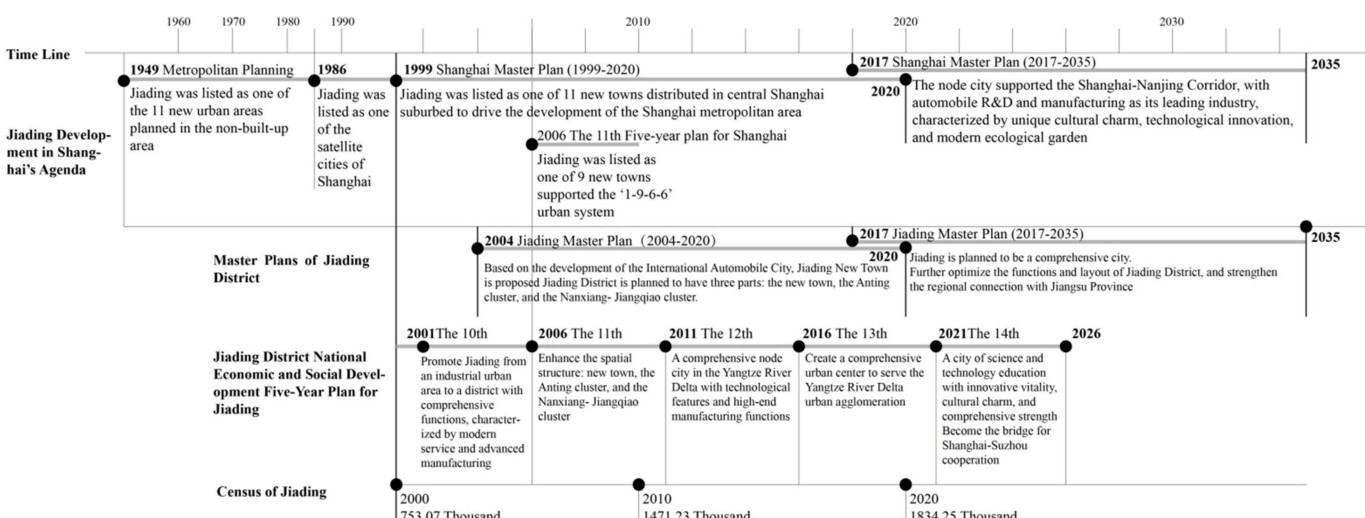

**Figure 2.** The development trajectory of Jiading District. Source: Authors' construction based on administrative documents from open resources. https://www.shanghai.gov.cn/ (accessed on 15 September 2023); http://www.jiading.gov.cn (accessed on 15 September 2023).

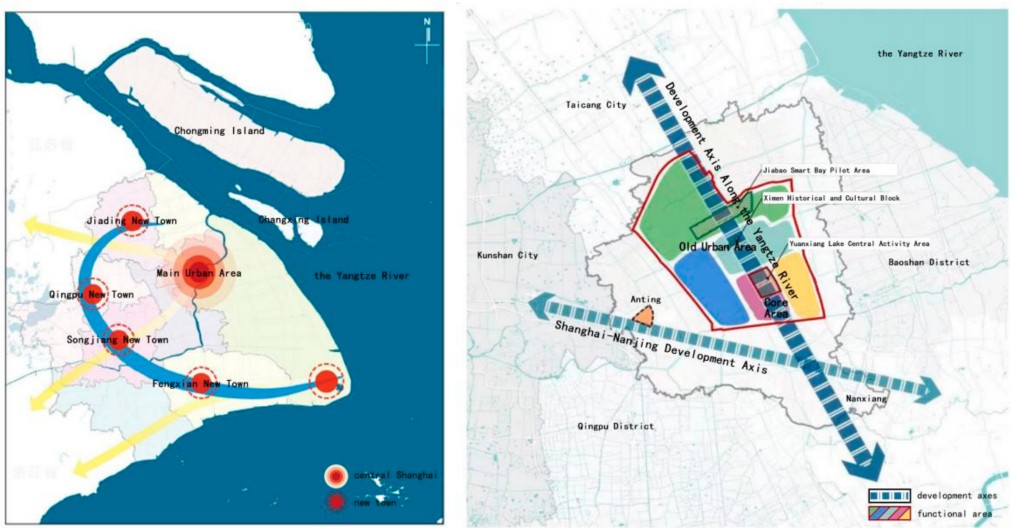

**Figure 3.** Location of Jiading. Source: Opinions on the implementation of the '14th Five-Year Plan' of Shanghai to accelerate the planning and construction of new towns (self-translated).

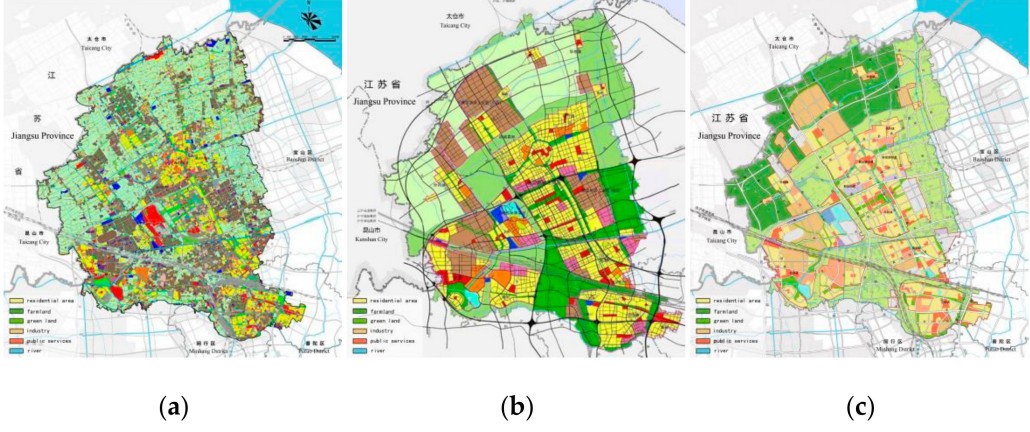

(**a**)　　　　　　　　(**b**)　　　　　　　　(**c**)

**Figure 4.** Land use status (**a**), Master plan (2010) (**b**), and Master plan (2017–2035) (**c**). Source: Jiading Master Plan from open resources. http://www.jiading.gov.cn (accessed on 15 September 2023).

Jiading was planned as one of the first five satellite cities, according to the 'Initial Opinions on Master Plan of Shanghai' issued in 1959. Its aim was to accommodate industries and the resident population of inner-city areas. However, because of a lack of development for living facilities, satellite cities developed slowly [31]. Successively, the 'Master Plan of Shanghai (1999–2020)' replaced the concept of a 'satellite city' with the new idea of new town/city development, aiming to support the development of Shanghai in suburbs, which took comprehensive functions as the new priority. Jiading New Town, listed as one of the nine New Towns of Shanghai in the 'Eleventh Five-Year Plan' period (2006–2010), was officially initiated on 18 August 2004, led by Jiading New Town Development Co., Ltd. Jiading District was, therefore, mainly composed of three parts, including the core area of Jiading New Town (i.e., the new urban center), Nanxiang Area, and Anting Area. In the same light, the 'Twelfth Five-Year Plan' emphasized the comprehensive function of Jiading and suggested an integrative development of both industries that provide job opportunities and towns that support cozy accommodations for everyday lives. Continually, in the 'Master Plan of Shanghai (2017–2035)' (Figure 4c), new towns become critical strategic spatial nodes that support the function of global city development and integrative development of the Yangtze River Delta. Officially issued in September 2022, the 'Spatial Cooperative Plan of Greater Shanghai Metropolitan Area' identified Jiading as the 'global function nodes' in the Yangtze River Delta urban agglomeration. Jiading is defined as an 'independent comprehensive node city' now, and its development is said to be towards more comprehensive and regional functions, serving as an urban and regional node.

In the development of Jiading, resident population scale changes and distribution have always been critical outcomes affecting spatial intervention and planning practice strategies. By the end of 2022, the area of Jiading was 463.16 square kilometers [32], whereas the scale of resident population increase over the past 20 years has been drastic. Led by a series of development strategies, its resident population has increased significantly from 753,070 in 2000 to 1,834,250 in 2020, according to the national census (Table 1). The long-term residents of Jiading increased by almost 1.1 million. The inflow of the resident population increased from 42% to 56% from 2000 to 2010. The resident population scale has achieved the development aim, and there has been a steady increase in the resident population moving in from outside of Shanghai. The proportion of the labor population has also been increasing steadily for the past 20 years, reaching 90.2% in 2020. The urbanization rate of Jiading is 84.5%, which is nearly 5% lower than that of 89.3% of Shanghai City.

**Table 1.** Census of Jiading New Town (long-term residents).

| National Census | Total (Thousand) | In-Floating Population Outside of Shanghai (Thousand) | Urban Population (Thousand) | Rural Population (Thousand) | Household (Thousand) | Household Size (People per Household) | Age Structure |
|---|---|---|---|---|---|---|---|
| Fifth (11–2000) | 753.07 | 213.55 | No data | No data | 241.34 | 2.80 | 0–14 (11.98%) 15–64 (78.07%) Above 65 (9.95%) |
| Sixth (11–2010) | 1471.23 | 828.19 | No data | No data | 542.67 | 2.37 | 0–14 (8.47%) 15–64 (84.32%) Above 65 (7.21%) |
| Seventh (11–2020) | 1834.25 | 1036.92 | 1550.38 | 283.87 | 730.64 | 2.27 | 0–14 (9.8%) 15–64 (90.2%) Above 60 (12.1%) |

Source: Authors' construction based on data from the Jiading Bureau of Statistics, http://www.jiading.gov.cn/tongji/publicity (accessed on 24 December 2020).

Comparing the plans and census of Jiading, it seems that the resident population scale has reached its goals after over 80 years of development. Nonetheless, Jiading

development did not ideally follow the planned path to attract the inner-city population of Shanghai [33–35]. Jiading was not well accepted by the inner-city residents, who did not recognize the new town as a part of Shanghai, while the in-flowed population from outside of Shanghai also felt it was hard to blend in. The increased residents, therefore, showed an obvious pattern in spatial distribution: they mainly settled in the city center or the peripheries of the new towns [36]. The spatial heterogeneity of resident population distribution is pronounced as the population increases.

*3.2. Population Data and LMU-Related Factors*

The resident population census of Jiading is available in 2010 and 2020, and they are used to compare the 10-year change. Except for the census data, we adopted a *WorldPop* dataset [37] to check a more detailed distribution of the residents at a resolution of 100 m. According to both datasets, Table 2 shows the total difference in the resident population of Jiading. Although the absolute resident population of *WorldPop* is lower than the census data, the difference between both datasets is quite close ('0.3631' and '0.3918' million). Therefore, we consider that the *WorldPop* data can reflect the trend of the resident population change in Jiading.

**Table 2.** Resident population data from two sources in Jiading (unit: million).

|  | **2010** | **2020** | **Difference** |
| --- | --- | --- | --- |
| The census | 1.4712 | 1.8343 | 0.3631 |
| WorldPop | 1.0927 | 1.4845 | 0.3918 |

Source: Authors' construction.

Figure 5 visualizes the resident population data at 100 m resolution in 2010 and 2020. It reveals that though the total population increased during the 10 years, residential areas of high density lie only in several parts. According to their differences, we can find that the population increases in most regions, while the green patches represent the decreased area. Especially, the largest influx occurs in the southeast corner since this part is close to the city center of Shanghai.

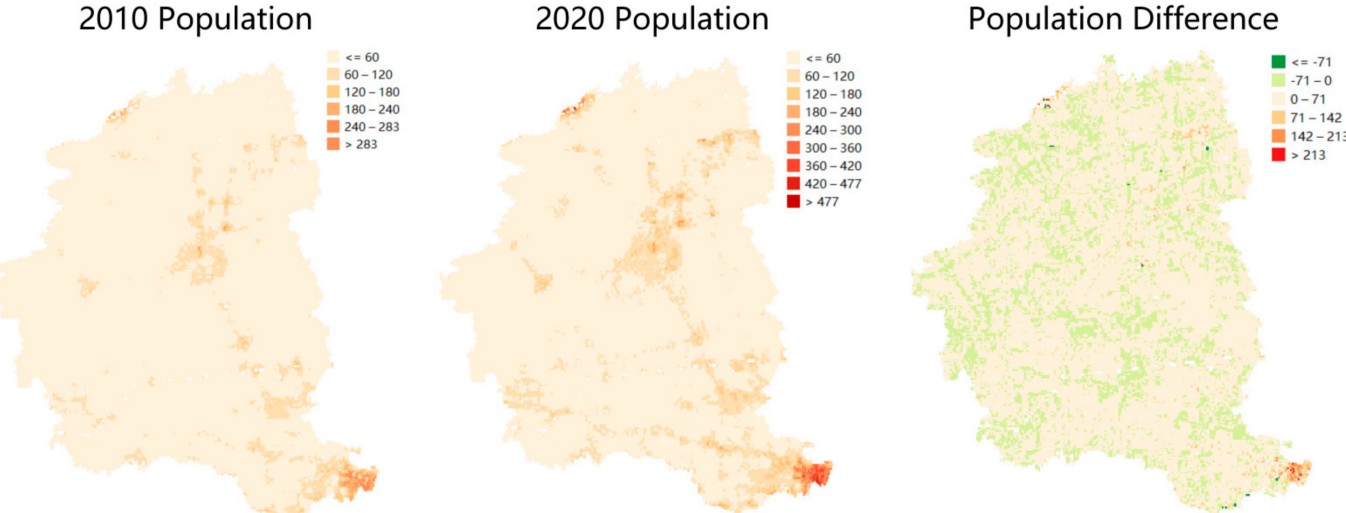

**Figure 5.** Change in resident population between 2010 and 2020 (100 m resolution) and the distribution of their difference. Source: Authors' construction.

As WorldPop data are represented in regular grid models (Figure 6), we set the grid granularity from 600 m to 3000 m with an increment of 300 m to investigate the scale effect of MAUP. We selected the basic resolution of 600 m since: (1) the resolution of population data (i.e., Worldpop) is $100 \times 100$ m; and (2) a ten-minute walking radius relates to 600 m. As a result, population data are aggregated for different resolutions (Figure 6). Meanwhile,

the data on resident population differences between 2010 and 2020 were aggregated into the nine resolutions as well. As involving fewer than 100 grids in Jiading, we selected 3000 m as the upper bound to avoid unreasonable high correlations.

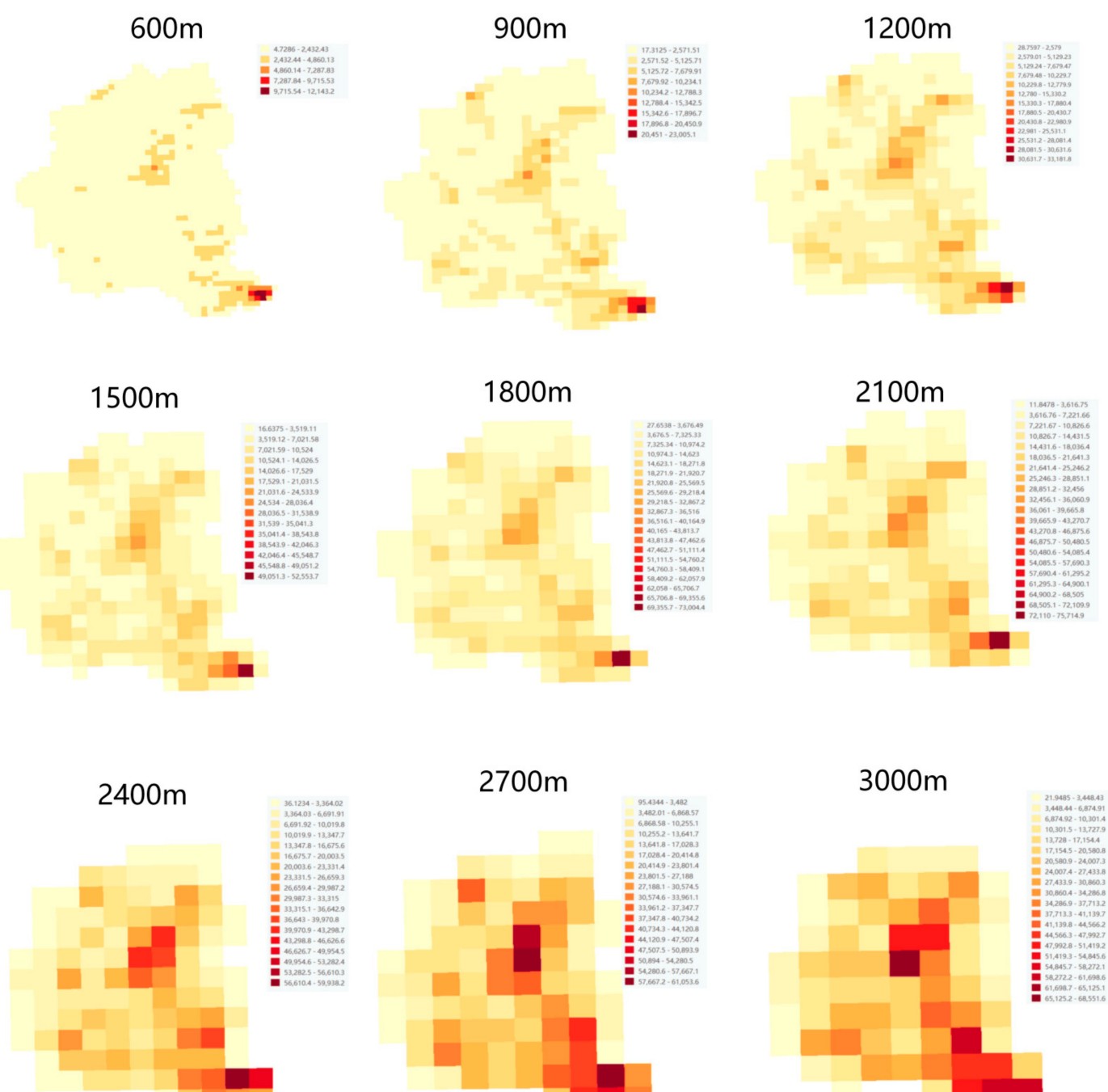

**Figure 6.** Different scales of resident population grids. Source: Authors' construction.

Moreover, PoI data are used to represent LMU factors. We transformed PoI data into point data first (Figure 7). All the PoIs were cleaned before the transformation. 'Metro Station' and 'Bus stop' reflect the transportation nodes in Jiading. 'Enterprise' and 'Shopping mall' present the referring locations of enterprises and shopping malls. 'Park & Square' represents the locations of a park or square for recreational activities. 'School' PoIs include primary and middle schools. 'Hospital' includes general hospitals, specialized hospitals, and township hospitals.

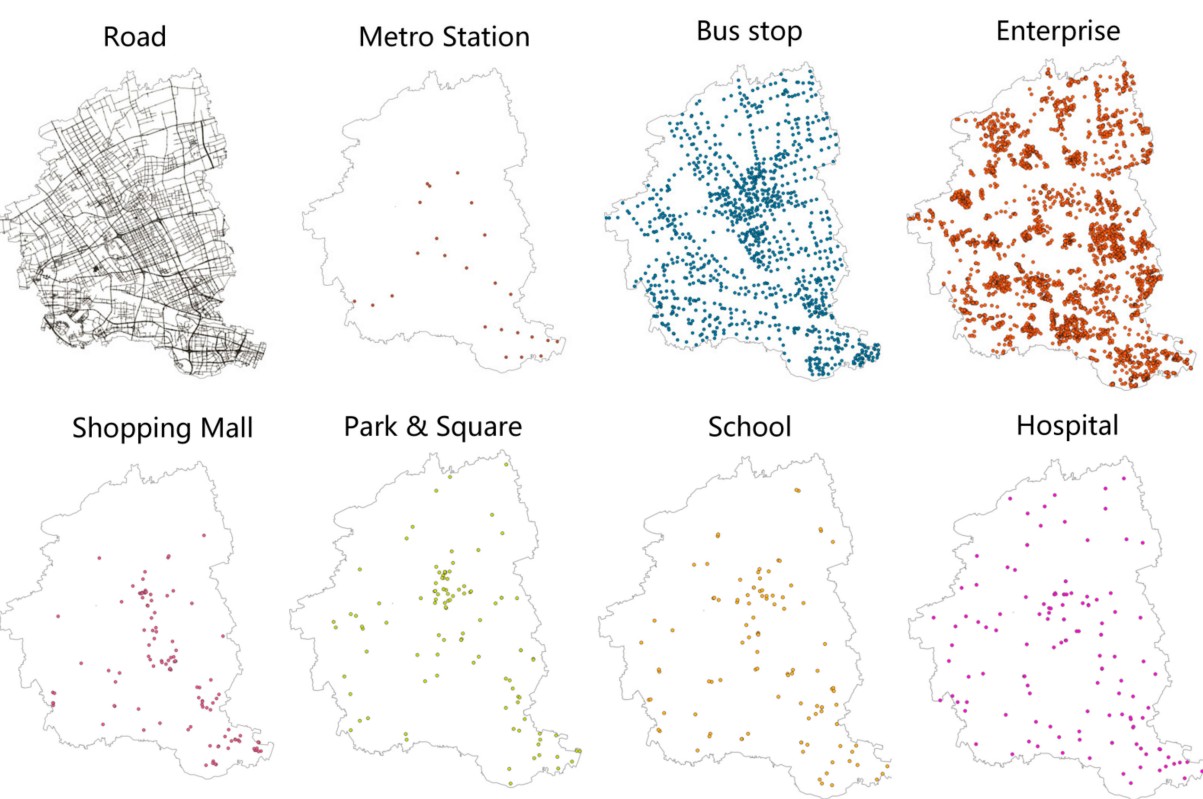

**Figure 7.** Open street map road data and different types of PoI. Source: Authors' construction.

To facilitate the computation of GD, we designed 14 independent variables (i.e., factors; see Table 3). Based on one type of PoI, grid attributes can be categorized according to the number of PoIs in a grid. For the accessibility of the grid to PoIs, we consider the minimum Euclidean distance between the PoIs and the centroid of a grid. We created the 14 factors in grid models that cover all these scales (600–3000 m).

**Table 3.** Indications of all independent variables.

| Type | Variable (Abb.) | Meaning |
|---|---|---|
| | RL | Road length |
| | BS_N | Bus stops |
| | SM_N | Shopping malls |
| The total number (N) in a grid | E_N | Enterprises |
| | S_N | Schools |
| | H_N | Hospitals |
| | PS_N | Parks and squares |
| | MS_MD | To metro stations |
| | BS_MD | To bus stops |
| | SM_MD | To shopping malls |
| Minimum (Euclidean) distance (MD) from a grid center | E_MD | To enterprises |
| | H_MD | To hospitals |
| | S_MD | To schools |
| | PS_MD | To parks and squares |

Source: Authors' construction.

Especially, metro stations are relatively sparse in Jiading, so we did not select the number of metro stations as a factor. To consider the influence of the subway, we designed the variable MS_MD to check the accessibility of a grid to the closest metro station (Table 3).

### 3.3. Relationships between the Resident Population and LMU Factors

We employed the GD to distinguish the key drivers for the distribution of the 2020 resident population and 10-year population change in Jiading. As independent variables only refer to discrete values in the GD, we adopted Jenks Natural Breaks Classification and zoned the attribute values of grids for each factor.

After running the GD computation for all the scales from 600 m to 3000 m, we obtained the results of its factor detector and interaction detector. The factor detector supports us in distinguishing these factors with the largest contribution to resident population distribution and its change, while the interaction detector can reveal the pairwise factors involving the most enhancement via interaction. We elaborate on them in the following parts.

#### 3.3.1. Resident Population Distribution (Y1) and Each LMU Factor

To ensure statistical validity, we checked the *p*-value of the results of the factor detector. For Y1 (2020 population) and Y2 (resident population difference between 2010 and 2020), we list all the values of q statistics and their *p*-values in Tables 4 and 5.

**Table 4.** The q-values of all the factors regarding the 2020 resident population (Y1).

| Factor | 600 m | | 900 m | | 1200 m | | 1500 m | | 1800 m | | 2100 m | | 2400 m | | 2700 m | | 3000 m | |
|---|---|---|---|---|---|---|---|---|---|---|---|---|---|---|---|---|---|---|
| | q | *p*-Value | q | *p*-Value | q | *p*-Value | q | *p*-Value | q | *p*-Value | q | *p*-Value | q | *p*-Value | q | *p*-Value | q | *p*-Value |
| **RL** | 0.14 | 0.00 | 0.19 | 0.00 | 0.24 | 0.00 | 0.30 | 0.00 | 0.30 | 0.00 | 0.37 | 0.00 | 0.39 | 0.00 | 0.41 | 0.00 | 0.43 | 0.00 |
| **BS_N** | 0.15 | 0.00 | 0.27 | 0.00 | 0.35 | 0.00 | 0.37 | 0.00 | 0.39 | 0.00 | 0.46 | 0.00 | 0.58 | 0.00 | 0.57 | 0.00 | 0.55 | 0.00 |
| **E_N** | 0.04 | 0.00 | 0.06 | 0.00 | 0.10 | 0.00 | 0.14 | 0.00 | 0.15 | 0.00 | 0.24 | 0.00 | 0.25 | 0.00 | 0.22 | 0.00 | 0.25 | 0.00 |
| **SM_N** | 0.06 | 0.00 | 0.15 | 0.00 | 0.14 | 0.01 | 0.27 | 0.00 | 0.36 | 0.00 | 0.37 | 0.00 | 0.44 | 0.00 | 0.27 | 0.02 | 0.42 | 0.01 |
| **S_N** | 0.01 | 0.11 | 0.03 | 0.05 | 0.09 | 0.00 | 0.18 | 0.00 | 0.21 | 0.00 | 0.28 | 0.00 | 0.35 | 0.00 | 0.35 | 0.00 | 0.43 | 0.00 |
| **H_N** | 0.01 | 0.18 | 0.05 | 0.01 | 0.12 | 0.00 | 0.14 | 0.00 | 0.19 | 0.00 | 0.20 | 0.00 | 0.33 | 0.00 | 0.29 | 0.01 | 0.43 | 0.02 |
| **PS_N** | 0.02 | 0.01 | 0.06 | 0.00 | 0.05 | 0.24 | 0.10 | 0.18 | 0.08 | 0.40 | 0.21 | 0.02 | 0.27 | 0.06 | 0.46 | 0.00 | 0.27 | 0.12 |
| **MS_MD** | 0.18 | 0.00 | 0.21 | 0.00 | 0.25 | 0.00 | 0.23 | 0.00 | 0.28 | 0.00 | 0.35 | 0.00 | 0.41 | 0.00 | 0.46 | 0.00 | 0.43 | 0.00 |
| **BS_MD** | 0.14 | 0.00 | 0.15 | 0.00 | 0.20 | 0.00 | 0.18 | 0.00 | 0.16 | 0.00 | 0.28 | 0.00 | 0.34 | 0.00 | 0.31 | 0.00 | 0.33 | 0.00 |
| **SM_MD** | 0.21 | 0.00 | 0.24 | 0.00 | 0.28 | 0.00 | 0.29 | 0.00 | 0.31 | 0.00 | 0.37 | 0.00 | 0.47 | 0.00 | 0.45 | 0.00 | 0.42 | 0.00 |
| **E_MD** | 0.07 | 0.00 | 0.09 | 0.00 | 0.13 | 0.00 | 0.15 | 0.00 | 0.14 | 0.00 | 0.22 | 0.00 | 0.24 | 0.00 | 0.31 | 0.00 | 0.22 | 0.01 |
| **H_MD** | 0.10 | 0.00 | 0.12 | 0.00 | 0.15 | 0.00 | 0.15 | 0.00 | 0.17 | 0.00 | 0.27 | 0.00 | 0.22 | 0.00 | 0.24 | 0.00 | 0.32 | 0.00 |
| **S_MD** | 0.16 | 0.00 | 0.19 | 0.00 | 0.23 | 0.00 | 0.20 | 0.00 | 0.26 | 0.00 | 0.32 | 0.00 | 0.37 | 0.00 | 0.40 | 0.00 | 0.36 | 0.00 |
| **PS_MD** | 0.11 | 0.00 | 0.15 | 0.00 | 0.16 | 0.00 | 0.16 | 0.00 | 0.16 | 0.00 | 0.23 | 0.00 | 0.23 | 0.00 | 0.22 | 0.00 | 0.31 | 0.00 |

Source: Authors' construction.

**Table 5.** The q-values of all the factors regarding resident population differences between 2010 and 2020 (Y2).

| Factor | 600 m | | 900 m | | 1200 m | | 1500 m | | 1800 m | | 2100 m | | 2400 m | | 2700 m | | 3000 m | |
|---|---|---|---|---|---|---|---|---|---|---|---|---|---|---|---|---|---|---|
| | q | *p*-Value | q | *p*-Value | q | *p*-Value | q | *p*-Value | q | *p*-Value | q | *p*-Value | q | *p*-Value | q | *p*-Value | q | *p*-Value |
| **RL** | 0.08 | 0.00 | 0.12 | 0.00 | 0.17 | 0.00 | 0.24 | 0.00 | 0.26 | 0.00 | 0.34 | 0.00 | 0.32 | 0.00 | 0.35 | 0.00 | 0.36 | 0.00 |
| **BS_N** | 0.08 | 0.00 | 0.17 | 0.00 | 0.25 | 0.00 | 0.34 | 0.00 | 0.30 | 0.00 | 0.38 | 0.00 | 0.51 | 0.00 | 0.46 | 0.00 | 0.53 | 0.00 |
| **E_N** | 0.05 | 0.00 | 0.08 | 0.00 | 0.15 | 0.00 | 0.16 | 0.00 | 0.17 | 0.00 | 0.27 | 0.00 | 0.31 | 0.00 | 0.26 | 0.00 | 0.27 | 0.00 |
| **SM_N** | 0.04 | 0.00 | 0.12 | 0.00 | 0.12 | 0.01 | 0.20 | 0.00 | 0.29 | 0.01 | 0.28 | 0.00 | 0.39 | 0.00 | 0.18 | 0.09 | 0.36 | 0.01 |
| **S_N** | 0.00 | 0.34 | 0.01 | 0.33 | 0.04 | 0.04 | 0.13 | 0.00 | 0.17 | 0.00 | 0.18 | 0.00 | 0.24 | 0.04 | 0.28 | 0.02 | 0.35 | 0.00 |
| **H_N** | 0.01 | 0.40 | 0.02 | 0.09 | 0.07 | 0.00 | 0.09 | 0.01 | 0.16 | 0.00 | 0.18 | 0.00 | 0.24 | 0.00 | 0.25 | 0.02 | 0.37 | 0.06 |
| **PS_N** | 0.01 | 0.03 | 0.03 | 0.03 | 0.03 | 0.40 | 0.06 | 0.36 | 0.05 | 0.55 | 0.12 | 0.13 | 0.18 | 0.17 | 0.32 | 0.07 | 0.22 | 0.25 |
| **MS_MD** | 0.10 | 0.00 | 0.13 | 0.00 | 0.15 | 0.00 | 0.17 | 0.00 | 0.21 | 0.00 | 0.27 | 0.00 | 0.30 | 0.00 | 0.43 | 0.00 | 0.40 | 0.00 |
| **BS_MD** | 0.09 | 0.00 | 0.10 | 0.00 | 0.15 | 0.00 | 0.15 | 0.00 | 0.13 | 0.00 | 0.27 | 0.00 | 0.32 | 0.00 | 0.27 | 0.00 | 0.35 | 0.00 |
| **SM_MD** | 0.12 | 0.00 | 0.16 | 0.00 | 0.18 | 0.00 | 0.23 | 0.00 | 0.24 | 0.00 | 0.34 | 0.00 | 0.42 | 0.00 | 0.42 | 0.00 | 0.41 | 0.00 |
| **E_MD** | 0.06 | 0.00 | 0.08 | 0.00 | 0.12 | 0.00 | 0.13 | 0.00 | 0.13 | 0.00 | 0.19 | 0.00 | 0.22 | 0.00 | 0.31 | 0.00 | 0.26 | 0.00 |
| **H_MD** | 0.06 | 0.00 | 0.09 | 0.00 | 0.12 | 0.00 | 0.13 | 0.00 | 0.15 | 0.00 | 0.26 | 0.00 | 0.20 | 0.00 | 0.23 | 0.00 | 0.29 | 0.00 |
| **S_MD** | 0.09 | 0.00 | 0.11 | 0.00 | 0.13 | 0.00 | 0.14 | 0.00 | 0.21 | 0.00 | 0.24 | 0.00 | 0.30 | 0.00 | 0.35 | 0.00 | 0.33 | 0.00 |
| **PS_MD** | 0.05 | 0.00 | 0.09 | 0.00 | 0.10 | 0.00 | 0.11 | 0.00 | 0.10 | 0.00 | 0.18 | 0.00 | 0.14 | 0.01 | 0.18 | 0.00 | 0.27 | 0.00 |

Source: Authors' construction.

In addition, we regard the largest q statistic of each scale as a benchmark and calculate the q-ratios of the other factors. These ratios are shown in Figure 8. They provide a view of the relative importance of each factor compared to the one with the largest q. In this sense, we can identify the main cross-scale drivers for the 2020 population.

According to Table 4, the most important variables for Y1 are the total number of bus stops, minimum distance to shopping malls and metro stations, road length, and minimum distance to schools (BS_N, SM_MD, MS_MD, and RL and S_MD), since they are significant on all the scales (i.e., crossing scales). Starting from 900 m, the total number of bus stops (BS_N) is the first factor interpreting the 2020 resident population distribution, which indicates interior public transportation largely contributes to this population distribution. The minimum distance to shopping malls (SM_MD) reflects that residents prefer the closeness and convenience of shopping malls. The minimum distance to metro stations (MS_MD) also involves a good explanatory power to Y1 and indicates local people's

preference for and reliance on subway transportation. Meanwhile, road length (RL) in the grids positively influences the related population to a large degree. At last, the minimum distance to schools (S_MD) covers relatively high q-ratios (>50%) in all the selected scales (Figure 8), and stably ranks in the middle (see Table 4). Thus, S_MD has a moderate explanatory power of Y1, and residents could give it high priority.

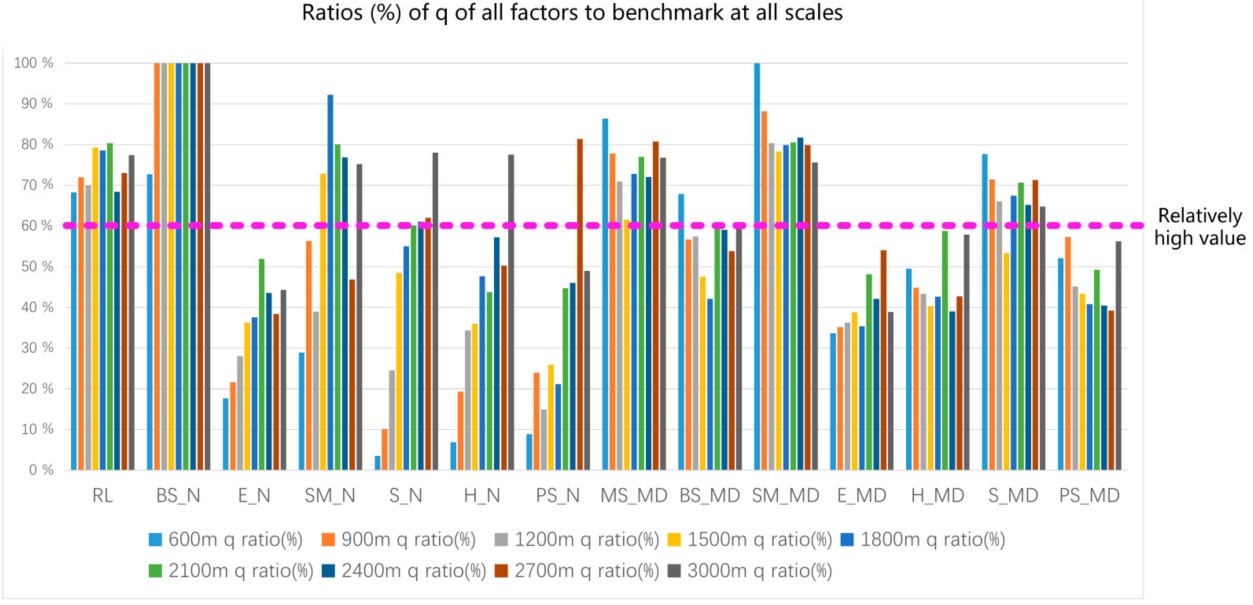

**Figure 8.** q-ratios of all the factors regarding the 2020 resident population (Y1). Source: Authors' construction.

In contrast, the other factors are either partially significant or insignificant to Y1. Both the total number of and minimum distance to enterprises (E_N and E_MD) are low-ranking factors (Table 4), and their q-ratios are not high (Figure 8). This result reveals enterprises have not much direct correlation to the 2020 resident population distribution, neither in number nor their minimum distances. The total number of shopping malls (SM_N) shows a high explanatory power for Y1 between the scales of 1500 m and 2400 m, which refers to its proper scales for interpreting Y1.

The total number of schools (S_N) is not statistically significant at 600 m and 900 m, and its q-ratios monotonously increase from 1200 m to 3000 m (Figure 8). Accordingly, we consider S_N to be more suitable for the discussion of the population distribution on larger scales (e.g., ≥3000 m), but not for the analysis of neighborhood levels in this paper. The test result of the total number of hospitals (H_N) is similar to that of S_N (Figure 8). In general, we believe S_N and H_N are not appropriate to be employed independently on scales smaller than 3000 m. The total number of parks and squares (PS_N) is not statistically significant on five scales, and its low ranks and q-ratios show PS_N is insignificant for interpreting Y1. The minimum distance to parks and squares (PS_MD), another factor about parks and squares, shows a general but not a notable correlation with Y1. Thus, we consider parks and squares less important for Y1 among the scales from 600 m to 3000 m.

Compared with BS_N, the factor BS_MD shows a moderate influence on Y1 (Figure 8). It implies that the density of bus stops in a grid is more important than the minimum distance to them. S_MD also presents moderate explanatory power to Y1, and its ranks and q-ratios are stable. So schools on these selected scales can firmly influence Y1 via its minimum distance. In contrast, the minimum distance to hospitals (H_MD) has a limited impact on Y1 and is hardly regarded as a key driver.

Moreover, we visualize the ranks of all these factors according to their q-values at each scale (Figure 9). In a word, among the selected scales, public transportation (BS_N, MS_MD, and RL) significantly shaped the 2020 resident population of Jiading. Additionally, the minimum distance to public services, including shopping malls and schools (SM_MD

and S_MD), is another group of key drivers of Y1. It implies residents are sensitive to the distance of their daily needs for shopping and school commuting.

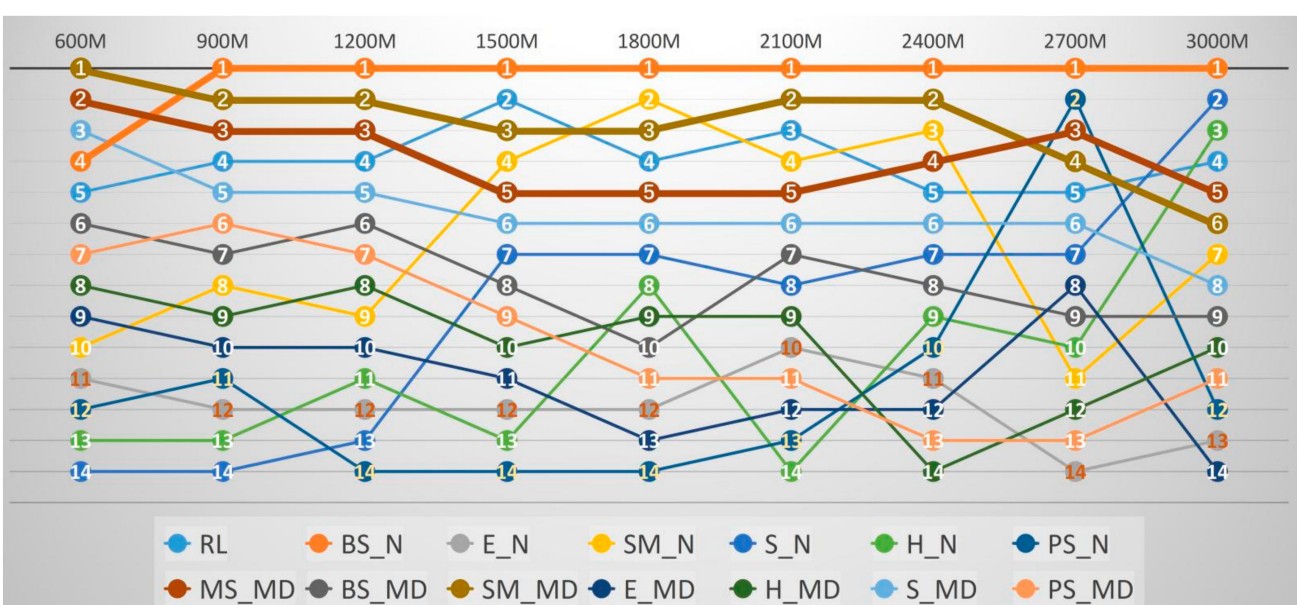

**Figure 9.** Rank q-values of all factors for the 2020 resident population (Y1). Source: Authors' construction.

### 3.3.2. Resident Population Change (Y2) and Each LMU Factor

Similarly, we conducted the same computation for the resident population difference (Y2) between 2010 and 2020. Table 5 and Figure 10 present the q statistic of all factors and their q-ratios, respectively. Similar to Figure 8, the largest q at each spatial scale is considered a benchmark, and we calculate the ratios of the other factors with their q-values. The rank of factors on each scale is visualized in Figure 11. Although it is difficult to elicit a cross-scale factor to interpret the phenomenon of resident population change, Figure 11 presents three key factors, i.e., the total number of bus stops, road length, and the minimum distance to shopping malls (BS_N, RL, and SM_MD). BS_N is the dominant driving factor for Y2. Its q-values are advantageously higher than those of the others since 900 m (Figure 11). Compared to BS_N, the minimum distance to bus stops (BS_MD) has less explanatory power than Y2. Similar to BS_MD, the minimum distance to schools (S_MD) has only moderate explanatory power for Y2.

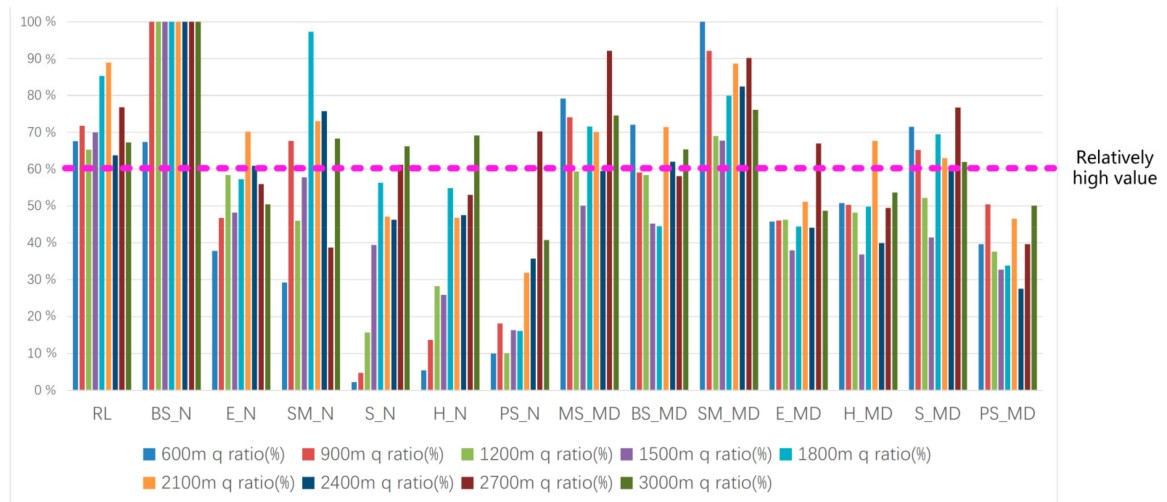

**Figure 10.** q-ratios of all factors regarding the 10-year resident population difference (Y2). Source: Authors' construction.

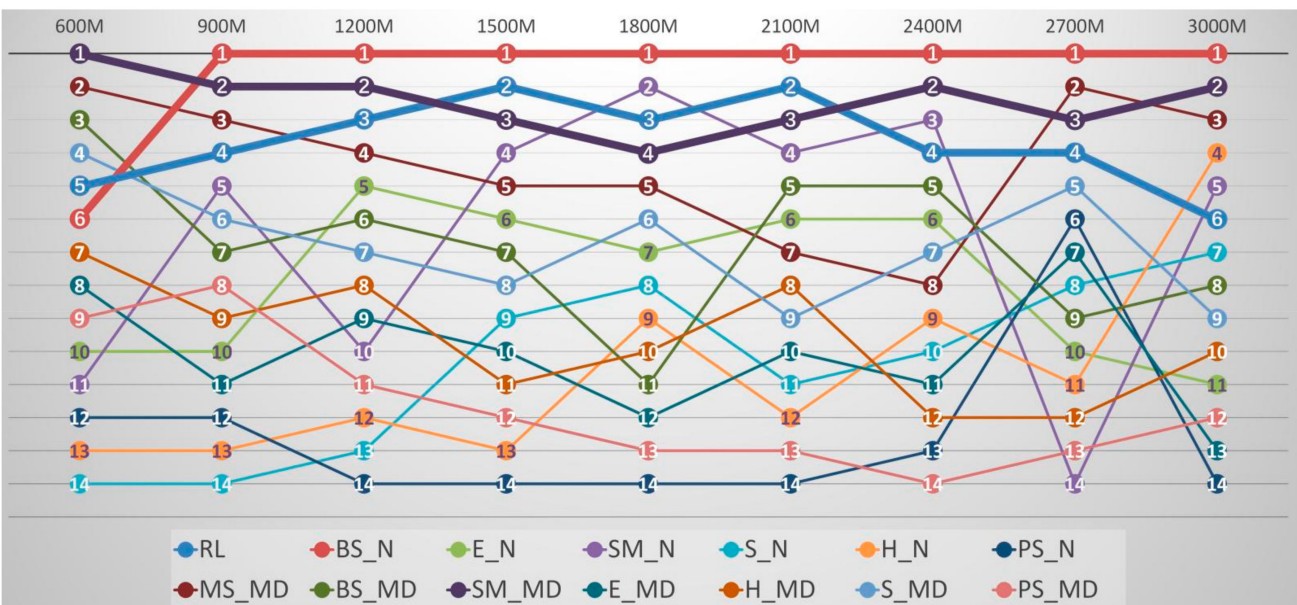

**Figure 11.** Rank q-values of all factors for the 10-year resident population difference (Y2). Source: Authors' construction.

The q-values of the other factors fluctuate on different scales. The minimum distance to metro stops (MS_MD) has obvious explanatory power for Y2 on most scales (see Figures 10 and 11), which means its use shall be considered for resident population change at neighborhood scales. The total number of shopping malls (SM_N) involves a significant explanatory power to Y2 from 1500 m to 2400 m. In addition, the total number of enterprises (E_N) presents a moderate influence on Y2 on the scales from 1200 m to 2400 m, while E_MD includes less importance for Y2. Similar to E_MD, H_MD, and PS_MD have weak explanatory power compared to Y2.

The total number of schools, hospitals, and parks and squares (S_N, H_N, and PS_N, respectively) all involve certain statistically insignificant scales, and their contribution to Y2 is limited (Figure 11). Thus, we consider the three factors that can be excluded from the discussion on Y2 below the scale of 3000 m.

### 3.3.3. Resident Population Distribution (Y1), Change (Y2), and Pairwise LMU Factors

After ranking the contribution of every independent factor to Y1 and Y2, we present the non-linear enhancement results of the interaction detector for Y1 and Y2 in Tables 6 and 7 and remove statistically insignificant results (e.g., those regarding PS_N at the scale of 1800 m). Non-linear enhancement stands for the most enhancement of factor interaction, i.e., the case 'q(x1 ∩ x2) > q(x1) + q(x2)' [25].

Furthermore, based on the results of the GD interaction detector, we confirmed the optimal scale for analyzing pairwise factors of resident population distribution/change in Jiading. For Y1, there is no more non-linear enhancement after the scale of 1800 m. It implies the q-values of the less important factors have largely increased since the scale of 2100 m. To avoid the 'scale effect' of MAUP, we consider sensitivities can be minimized at the 1800 m scale, and it is the optimal scale to distinguish the relative importance of the factors for Y1. For Y2, the scales of 2100 m and 2400 m both involve only one non-linear enhancement. In this case, we also consider the scale of 1800 m as the optimal one for analysis.

**Table 6.** Non-linear enhancements regarding the 2020 resident population (Y1).

| 600 m | | 900 m | | 1200 m | | 1500 m | | 1800 m | |
|---|---|---|---|---|---|---|---|---|---|
| **Interaction** | **q** | **Interaction** | **q** | **Interaction** | **q** | **Interaction** | **q** | **Interaction** | **q** |
| RL ∩ E_N | 0.18 | RL ∩ E_N | 0.27 | BS_N ∩ E_N | 0.48 | E_N ∩ SM_N | 0.47 | BS_N ∩ E_N | 0.68 |
| RL ∩ SM_N | 0.20 | BS_N ∩ E_N | 0.35 | E_N ∩ SM_N | 0.26 | E_N ∩ S_N | 0.36 | E_N ∩ SM_N | 0.52 |
| BS_N ∩ E_N | 0.20 | E_N ∩ SM_N | 0.25 | E_N ∩ S_N | 0.20 | E_N ∩ MS_MD | 0.38 | E_N ∩ S_N | 0.46 |
| E_N ∩ SM_N | 0.11 | E_N ∩ PS_N | 0.23 | E_N ∩ H_N | 0.26 | SM_N ∩ E_MD | 0.44 | E_N ∩ H_N | 0.35 |
| E_N ∩ H_MD | 0.15 | E_N ∩ H_MD | 0.22 | E_N ∩ PS_MD | 0.31 | S_N ∩ H_MD | 0.36 | E_N ∩ PS_MD | 0.46 |
| E_N ∩ S_MD | 0.20 | E_N ∩ PS_MD | 0.23 | SM_N ∩ E_MD | 0.28 | | | SM_N ∩ E_MD | 0.52 |
| E_N ∩ PS_MD | 0.16 | SM_N ∩ E_MD | 0.26 | | | | | S_N ∩ E_MD | 0.40 |
| PS_N ∩ E_MD | 0.09 | H_N ∩ E_MD | 0.15 | | | | | H_N ∩ E_MD | 0.34 |
| | | | | | | | | E_MD ∩ H_MD | 0.32 |
| | | | | | | | | E_MD ∩ PS_MD | 0.31 |

Source: Authors' construction.

**Table 7.** Non-linear enhancements regarding the 10-year population difference (Y2).

| 600 m | | 900 m | | 1200 m | | 1500 m | | 1800 m | |
|---|---|---|---|---|---|---|---|---|---|
| **Interaction** | **q** | **Interaction** | **q** | **Interaction** | **q** | **Interaction** | **q** | **Interaction** | **q** |
| RL ∩ SM_N | 0.13 | RL ∩ SM_N | 0.25 | RL ∩ SM_N | 0.29 | RL ∩ SM_N | 0.49 | BS_N ∩ E_N | 0.64 |
| BS_N ∩ E_N | 0.14 | BS_N ∩ E_N | 0.26 | BS_N ∩ E_N | 0.50 | E_N ∩ SM_N | 0.43 | E_N ∩ SM_N | 0.51 |
| E_N ∩ SM_N | 0.10 | E_N ∩ SM_N | 0.25 | E_N ∩ SM_N | 0.34 | E_N ∩ S_N | 0.34 | E_N ∩ S_N | 0.43 |
| E_N ∩ H_MD | 0.12 | E_N ∩ H_MD | 0.18 | E_N ∩ S_N | 0.20 | E_N ∩ MS_MD | 0.36 | E_N ∩ H_N | 0.35 |
| E_N ∩ S_MD | 0.14 | E_N ∩ PS_MD | 0.18 | E_N ∩ H_N | 0.26 | E_N ∩ S_MD | 0.31 | E_N ∩ PS_MD | 0.40 |
| E_N ∩ PS_MD | 0.11 | SM_N ∩ PS_N | 0.15 | E_N ∩ H_MD | 0.28 | E_N ∩ PS_MD | 0.29 | SM_N ∩ S_N | 0.47 |
| SM_N ∩ PS_N | 0.05 | SM_N ∩ E_MD | 0.24 | E_N ∩ PS_MD | 0.34 | SM_N ∩ E_MD | 0.34 | SM_N ∩ H_N | 0.48 |
| SM_N ∩ E_MD | 0.09 | | | SM_N ∩ S_N | 0.17 | S_N ∩ E_MD | 0.28 | SM_N ∩ E_MD | 0.48 |
| PS_N ∩ E_MD | 0.08 | | | SM_N ∩ H_N | 0.21 | S_N ∩ H_MD | 0.30 | S_N ∩ E_MD | 0.33 |
| PS_N ∩ H_MD | 0.08 | | | SM_N ∩ E_MD | 0.27 | MS_MD ∩ H_MD | 0.30 | H_N ∩ E_MD | 0.33 |
| E_MD ∩ H_MD | 0.13 | | | SM_N ∩ H_MD | 0.26 | | | E_MD ∩ H_MD | 0.31 |
| E_MD ∩ PS_MD | 0.11 | | | S_N ∩ E_MD | 0.17 | | | E_MD ∩ PS_MD | 0.26 |

| 2100 m | | 2400 m | |
|---|---|---|---|
| **Interaction** | **q** | **Interaction** | **q** |
| SM_N ∩ H_N | 0.48 | H_N ∩ E_MD | 0.49 |

Source: Authors' construction.

We only check non-linear enhancements since we aim to find potential key drivers of Y1/Y2. These enhancements represent the prominent combinations of two factors that greatly facilitate the explanatory power of Y1/Y2. In short, non-linear enhancements mostly relate to enterprises (see Table 6), e.g., E_N. Its prominent interactions with other factors include the following:

(1) The minimum distance to schools (S_MD) at 600 m scale;
(2) The road length (RL), number of bus stops (BS_N), and number of shopping malls (SM_N) at 900 m scale;
(3) BS_N and minimum distance to the parks and squares (PS_MD) at 1200 m scale;
(4) SM_N and the minimum distance to metro stations (MS_MD) at 1500 m scales;
(5) The number of bus stops, shopping malls, and schools, and the minimum distance to the parks and squares at 1800 m.

At different scales, E_N is non-linearly enhanced with the above factors. We found that E_N, together with these factors, can enhance its causal link to Y1 (between 900 m and 1800 m), though it is not a primary independent factor for Y1.

Specifically, the interacted q scores of BS_N ∩ E_N at 900 m, 1200 m, and 1800 m are the largest, and they could be an essential explanation for Y1 on these scales. This could result from transportation activities, such as commuting to work. On these micro-scales, the number of bus stops and enterprises forms an indication of resident population aggregation. The pair of E_N ∩ SM_N and that of SM_N ∩ E_MD both gain high q scores at 1500 m and 1800 m. This result reveals the notable combined influence on Y1 of life convenience (SM_N) and commuting to work (E_N and E_MD).

Table 7 lists the non-linear enhancement of two factors for Y2 on different scales. We also seek the key drivers for Y2. E_N is still a significant factor in non-linear enhancements to Y2. Its prominent interactions with other factors include the following:

(1) BS_N and S_MD at 600 m scale;
(2) BS_N, SM_N, and PS_MD at 900 m scale;
(3) BS_N, SM_N, and PS_MD at 1200 m scale;
(4) SM_N, MS_MD, and S_N at 1500 m scale;
(5) BS_N, SM_N, S_N, and PS_MD at 1800 m scale.

BS_N∩E_N (from 600 m to 1200 m and 1800 m) involves a prominent interaction. In addition, we can conclude that the interaction of E_N and SM_N mostly relates to high q scores. It indicates E_N and SM_N can considerably influence Y2 via factor interaction between the scales of 600 m and 1800 m.

A little surprisingly, the interaction of E_N and PS_MD (900 m, 1200 m and 1800 m) has great explanatory powers for Y2. Both PS_MD and E_N have weak explanatory power for Y2 (see Section 3.3.2), but their combination on specific scales presents an obvious influence on Y2. This result reveals the importance of work and recreational possibilities on neighborhood levels for residents.

### 3.3.4. Main Findings

Here, we summarize the findings of our test, especially on suitable scales of these factors to interpret the resident population distribution/change. We demonstrate the use of the proposed explanatory framework by introducing the findings.

(1) Main cross-scale drivers for 2020 resident population distribution (Y1).

The main cross-scale drivers for Y1 are summarized as follows:

- Transport convenience, including the number of bus stops, road length, and the minimum distance to metro stations (BS_N, RL, and MS_MD), significantly shapes Y1 from a grid scale of 600 m to 3000 m.
- Life convenience, including the minimum distance to shopping malls and schools (SM_MD and S_MD), maintains significant explanatory power for Y1 at all scales, while SM_SD was revealed to be more important.
- The number of enterprises (E_N) interacting with other factors (e.g., RL, BS_N, and SM_N) can considerably enhance its causal link to Y1, between the scales of 900 m and 1800 m.

In general, single spatial factors affect the population distribution, and the population changes differently. A special case is the number of enterprises (E_N). It reflects local job opportunities, and it showed a weak influence on the 2020 resident population distribution. However, it shows a crucial effect on the population distribution when interacting with the other factors (see Section 3.3.3).

(2) Key drivers of 10-year resident population change in Jiading (Y2).

Cross-scale key drivers for Y2 are summarized as follows:

- Single factors of the number of bus stops, road length, and the minimum distance to shopping malls (BS_N, RL, and SM_MD) can mainly interpret Y2 from the scale of 600 m to 3000 m.

- The numbers of enterprises and shopping malls (E_N, SM_N) present significant spatial correlations to Y2 by interacting with other factors between the scales of 600 m and 1800 m.

Similar to the case of Y1, resident population change influenced by LMU factors can be interpreted at different spatial scales with various combinations. For instance, the number of shopping malls (SM_N) presents its independent significance at the grid scale of 1500–2400 m, while its influence further increases via interacting with the following factors:

- Road length (RL) at 600 m;
- The numbers of enterprises (E_N) at 900 m and 1200 m;
- The road length at 1500 m (the most influential);
- The number of enterprises at 1800 m (the second influential).

(3)  The optimal scale for LMU factors of resident population change (Y2).

Figure 12 shows the influence of mixed functional land uses on population change. For Y2, we visualize q scores of non-linear interactions at scales from 600 m to 1800 m. Compared with the other four scales (600 m, 900 m, 1200 m, and 1500 m), the scale of 1800 m involves many pairs of factors that contain great explanatory powers. As mentioned before, we confirmed the optimal scale for analysis is 1800 m.

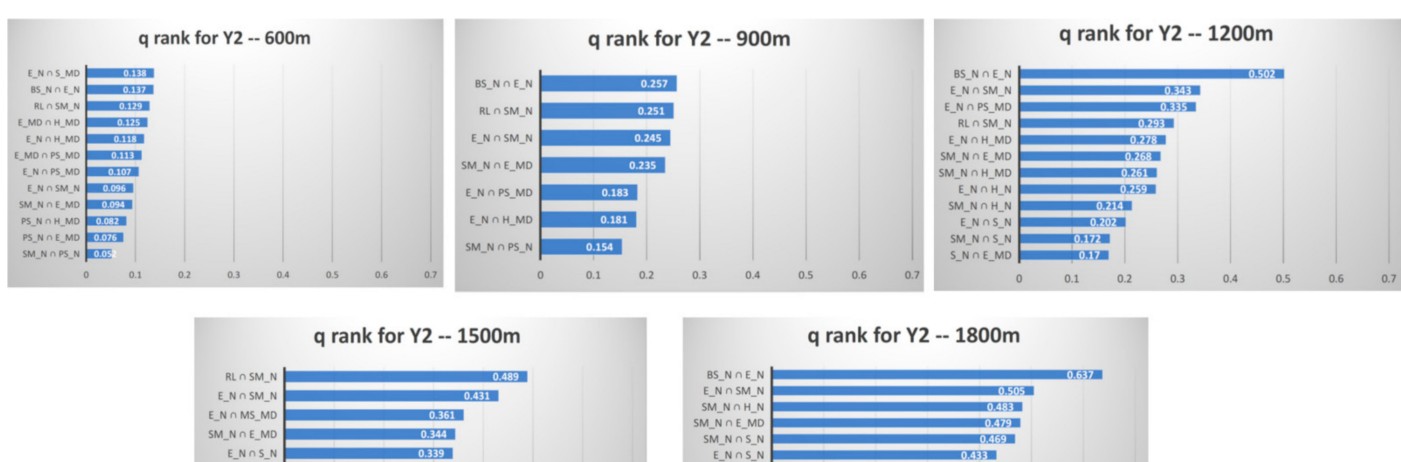

**Figure 12.** q scores of pairwise factor interactions for Y2 at scales between 600 m and 1800 m. Source: Authors' construction.

At 1800 m, the interaction of E_N and BS_N has the largest explanatory power for Y2 (q = 0.637). It means over 60% of their combined spatial distribution is consistent with the population change in Jiading between 2010 and 2020, which implies the configuration of enterprises coupled with bus stops, which provide convenient transport to the enterprises. We can infer that neighborhoods with more enterprises and bus stops involve a larger resident population. Shopping malls and enterprises in neighborhoods (1800 m) can also interpret over 50% of the 10-year resident population change (E_N ∩ SM_N).

Moreover, at 1800 m, the population change is sharply consistent with the interactions including SM_N + H_N (0.483), SM_N + E_MD (0.479), SM_N + S_N (0.469), E_N + S_N (0.433), E_N + PS_MD (0.4), E_N + H_N (0.353), S_N + E_MD (0.33), E_MD + H_N (0.327), and E_MD + H_MD (0.307). These interactions regarding different facilities prove the land-use mix has facilitated population change. In Jiading, we can expect more residents to be introduced to a neighborhood that includes plenty of easy-to-access enterprises, convenient bus transportation, enough shopping malls, hospitals, and schools, as well as short-distance parks and squares.

In addition to the optimal scale, there are some other worth-noticing findings. Similar to 1800 m, pairwise factors of BS_N and E_N refer to top ranks at 600 m, 900 m, and 1200 m. The pair of E_N and SM_N also shows strong explanatory power for Y2 at the scales of 900 m, 1200 m, and 1500 m. Road length (RL) is a significant factor below the scale of 1800 m. The interaction of RL and SM_N has high scores at 600 m (q = 0.129), 900 m (q = 0.251), 1200 m (q = 0.293), and 1500 m (q = 0.489). Thus, in the microscales below 1500 m, the accessibility to shopping malls is significant for population change. In the above cases, we reveal the applicable scales of different pairwise factors where these factors present great explanatory power.

## 4. Discussions

Drawing on the findings resulting from the explanatory framework applied in Jiading District, our hypothesis was verified to some extent.

(1) Reflecting the LMU outcome of Jiading by applying resident population distribution and changes, a set of 14 related factors is selected. They showed distinct effects at the grid scale from 600 m to 3000 m.
(2) The outcome shows that the optimal scale for this set of factors is 1800 m, approximately ten street blocks.
(3) It also helps identify the most influential factors at the grid scale of 1800 m and factors' interactions in pairs, which can provide more accurate support for spatial interventions, realizing efficient LMU for planners and policymakers.

Existing studies have shown that LMU significantly contributes to facilitating everyday lives [38–40], stimulating and diversifying economic dynamics [40,41], and improving the efficiency of land uses [42], which supports the idea that LMU is applied to reach specific development goals and fulfill the constantly evolving everyday needs of people [1]. Therefore, indexes and factors were different when they were selected to measure the outcomes of LMU and to guide the intervention based on these results [43]. For example, the criteria for new towns, old city centers, and areas aimed at encouraging industries have different recipes for LMU patterns [44], and so does the compatibility. Among all the effects of LMU, it has a specific impact on population distribution and social diversity [3,45,46]. Desk studies, empirical studies, and model analyses have been applied to reflect and reveal the in-depth interrelationships.

In this light, this study engaged in the discourse of LMU from the perspective of developing a novel explanatory framework to identify an optimal scale for a certain set of LMU factors and to reveal the most influential factors and their interactions at the identified scale. The framework is tested in the case of Jiading; in this case, it aims at a balanced resident population distribution. It could also be applied in different contexts with different sets of LMU factors aiming at distinct development goals. In this way, an appropriate scale for specific and context-related goals of LMU is suggested, and the most influential factors in single and pairwise forms are identified, supporting planners and policymakers to work on LMU approaches.

In addition, a previous study [29] for Wuhan city pointed out that population and economic activities involve obvious aggregation, while land use presents a decentralized state. The conclusion may overlook the different effects of these indicators at various spatial scales since they are not expected to functionalize in the same range on a neighborhood level.

In our case, some factors, such as bus stop number, showed a strong effect across all scales from 600 m to 3000 m in terms of affecting resident population distribution in 2020 and the population change between 2010 and 2020 in Jiading District. Some factors, such as the number of parks and squares, merely showed a strong effect at certain scales, which indicates that increasing the resident population distribution by enhancing LMU through adding parks and squares shows the best performance at a grid scale of 2700 m (approximately 10–11 blocks). It is noteworthy that some factors showed weak effects in a standalone way, while others presented a dominant impact by interacting with other factors. These types of factors were most likely to be overlooked in their impact on LMU.

For example, the interactions of enterprise numbers with other factors present significant spatial correlations to both the resident population distribution and population changes within the research area.

## 5. Conclusions

As already shown in the existing literature, increasing the LMU can stimulate urban dynamics [18,47], and the discussions were mainly centered around land use patterns of different types of land uses [12,48]; or avoiding NIMBY syndromes that suggested the integration of mutually reinforcing (land) functions [42,49]. In this study, we argue that these discussions cannot avoid the scale that is identified as one of the most influential elements in the discussion of LMU [9,19]. The LMU 'recipes' need to be studied more closely on multiple scales to guide spatial interventions and fill the gaps between theories and practices. Thus, it is also important to identify the effects of LMU-related factors at different scales. In this study, resident population distribution and 10-year population change were applied to reflect these effects.

We emphasized that scale plays a crucial role in affecting LMU implementation through the development of an explanatory framework. It helps to identify the most appropriate spatial scale for LMU with the related factors. It, therefore, supports more efficient planning strategies related to LMU. Furthermore, the study suggested an explanatory framework that opens to working with different sets of LMU-related factors in various contexts to identify the most appropriate spatial scale for planning interventions and efficient sets of factors. By working with pairwise factors, the framework also helps us to avoid overlooking the LMU-related factors that showed less power working alone but worked well with another factor. In our case, it shows that single factors such as the number of enterprises can be less interpretative for resident population distribution, which indicates that merely increasing job opportunities cannot increase residents while it becomes the critical one interacting with the others. Therefore, our framework provides alternative supports for decision making and policy making for urban–rural development, aiming at better efficiency of smart growth and sustainability through LMU. This also contributes to alternative approaches to promote the understanding of mixed-land-use practice beyond the case study region.

So far, we have succeeded in revealing the applicable scales and influence of pairwise factors to interpret the resident population distribution and changes in Jiading District with the identified factors within our framework. In the future, this study will proceed with applying this framework to different areas with various sets of LMU-related factors in distinct contexts. It could obtain a more in-depth understanding of the relationships between LMU and resident population distribution and changes.

**Author Contributions:** Conceptualization, L.L. and H.H.; Methodology, L.L. and H.H.; Validation, L.L. and H.H.; Formal analysis, L.L.; Investigation, L.L. and J.Q.; Resources, H.H. and J.Q.; Data curation, L.L.; Writing—original draft, L.L. and H.H.; Writing—review & editing, L.L. and H.H.; Visualization, L.L. and J.Q.; Supervision, H.H.; Funding acquisition, H.H. All authors have read and agreed to the published version of the manuscript.

**Funding:** The research has been funded by the National Key Research and Development Plan (Grant No. 2022YFC3800801), National Natural Science Foundation of China (NSFC, Grant No. 52208075), and the Shanghai Pujiang Program (Grant No. 2021PJC113). It was also supported by the Fundamental Research Funds for the Central Universities.

**Data Availability Statement:** The original contributions presented in the study are included in the article, further inquiries can be directed to the corresponding author.

**Acknowledgments:** Our acknowledgments go to Xinyun Tang. She supported the translation of 2 figures and collected general information of Jiading District at the initial stage.

**Conflicts of Interest:** The authors declare no conflict of interest.

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
