# Peer review of "Towards a Multi-Scale Effect of Land Mixed Use on Resident Population—A Novel Explanatory Framework of Interactive Spatial Factors"

_land, doi:10.3390/land13030331_

Round 1
Reviewer 1 Report
Comments and Suggestions for Authors
The article is overloaded with too many results of simple calculations of empirical authors data. The graphics and maps submitted for review are of poor quality and do not allow you to view important details of the images. The tables need to be simplified because there is no clear excess of data presented. The conclusions do not have proper specifics in the discussion. All graphic and tabular materials should be carefully reworked, and more understandable and non-abstract conclusions should be written.
Author Response
The main changes have been highlighted in the text.
The article is overloaded with too many results of simple calculations of empirical authors data. The graphics and maps submitted for review are of poor quality and do not allow you to view important details of the images. The tables need to be simplified because there is no clear excess of data presented. The conclusions do not have proper specifics in the discussion. All graphic and tabular materials should be carefully reworked, and more understandable and non-abstract conclusions should be written.
Response:
Thank you for your feedback on the manuscript. We appreciate your insights, which have helped us identify areas for improvement. We've made the following revisions based on your suggestions:
- We have replaced the vague Figures with higher quality versions (Fig. 3, 4, 5,6,7,8,10, and 12).
- We have simplified and modified lengthy tables to better present our test results, including Table 4,5,6, and 7.
- Following your advice, we have thoroughly rewritten the 'Discussions' and 'Conclusions' sections to enhance readability.
We believe these changes have significantly enhanced the clarity of the manuscript.
Reviewer 2 Report
Comments and Suggestions for Authors
Language revision is required as there are several instances throughout the text where the meaning is unclear due to poor sentence structure and/ or the term used is not adequate for the meaning that was intended. I had some problems following the line of argumentation. There are also some typos that any language revision software would have caught (i.e. catigories instead of categories; family sale instead of, I assume family scale, but the correct term would actually be household size; singly spatial, I guess single?).
Also, the visualizations have all very poor resolution which made it impossible to accompany whether the text and the visuals did indeed support each other. No clue if the text reports on findings adequately.
Other issues - the paper is reportedly about land-use mix and the ideal LUM at different scales to support economic growth. However it calculates quantity of uses and population. There is no data analyzed that would reflect economic growth and the land uses show little disaggregation in order to discusses mix of land uses for economic growth potential. I would recommend simplifying the contribution to focus on the exploration of the relationship between population and land-use mix - in order to also represent the types of land-uses that were possible to discern with the available data. Or, if the data allows it, labour and diversity of entreprises (maybe by size of employed labour force) to be also included?
In terms of population, please clarify if in population you consider resident population or workers. I would assume the first but distinguishing between population employed in the area, and residing in the area, might also contribute to explore the attraction of the area based on land-use mix.
Also, in the methodology adopted accessibility is a key component that influences the "attraction" of the scale of analysis in question. This issue of accessibility should be explored in the assumptions and in the discussion.
The authors use interchangeably the terms methodology and framework. These are not the same thing. Please choose where you are contributing (is it a methodology? Is it a framework?) and discuss how this contributes to the field. Be consistent with the use of the chosen term.
The findings reported in the discussion section are not novel or unexpected. That access to transportation nodes, life convenience, shopping centers etc positively correlates with population is a well known and researched link in both quantitative and qualitative or policy oriented studies. So I would like to see the authors reporting on what the methodology reported brought new to existing knowledge. A suggestion is to integrate the current discussion section with the reporting of the findings, and in the discussion explore of these findings mean in terms of policy guidelines of how best to optimize the LUM, accessibility, population distribution and change, etc.
I would also suggest taking a step back from the listing in the conclusions and critically reflect on what you find using your methodology. For example, that "some factors merely show a strong effect at certain scale" is a trivial finding. Of course the influence of parks and squares or of shopping malls will be more felt at a certain scale than at another. Different land-uses have different impacts, any urban planner would know this. But how can your methodology be used to optimize how to allocate these functions - so think at the level of how a planner or a policy maker could utilize your approach to optimize land-use allocation decisions.
Comments on the Quality of English Language
See above
Author Response
The main changes have been highlighted in the text.
Language revision is required as there are several instances throughout the text where the meaning is unclear due to poor sentence structure and/ or the term used is not adequate for the meaning that was intended. I had some problems following the line of argumentation. There are also some typos that any language revision software would have caught (i.e. catigories instead of categories; family sale instead of, I assume family scale, but the correct term would actually be household size; singly spatial, I guess single?).
Response:
Thank you for the insightful comments. We have diligently addressed the language issues throughout the manuscript. Additionally, we have rectified the previous typos and misapplications of terms, such as the correction of 'household size' in Table 1.
Also, the visualizations have all very poor resolution which made it impossible to accompany whether the text and the visuals did indeed support each other. No clue if the text reports on findings adequately.
Response:
Thank you for the feedback. Following your advice, we've replaced the vague Figures with higher quality versions (Fig. 3, 4, 5,6,7,8,10, and 12) in the revision.
Other issues - the paper is reportedly about land-use mix and the ideal LUM at different scales to support economic growth. However it calculates quantity of uses and population. There is no data analyzed that would reflect economic growth and the land uses show little disaggregation in order to discusses mix of land uses for economic growth potential. I would recommend simplifying the contribution to focus on the exploration of the relationship between population and land-use mix - in order to also represent the types of land-uses that were possible to discern with the available data. Or, if the data allows it, labour and diversity of entreprises (maybe by size of employed labour force) to be also included?
Response:
Thank you for your helpful suggestions. Indeed, this study primarily focuses on examining the multi-scale impact of land mixed uses (LMU) on resident population distribution and its change between 2010 and 2020. Following your advice, we have strengthened the related sections to enhance accuracy and focus. We consider resident population distribution as an outcome of LMU, and apply an explanatory framework to analyse the optimal spatial scale of a given set of LMU factors, along with their interaction effects on the resident population.
The discussion regarding the mix of land uses for economic growth potential is not within the scope of our study. Due to limitations in available datasets, we regretfully could not incorporate information on labor and the diversity of enterprises. We apologize for any misunderstanding this may have caused. In general, we have adopted your suggestions and modified our focus to emphasize LMUs and resident populations at various spatial scales.
In terms of population, please clarify if in population you consider resident population or workers. I would assume the first but distinguishing between population employed in the area, and residing in the area, might also contribute to explore the attraction of the area based on land-use mix.
Response:
Thanks again for the constructive suggestions. We found them to be very helpful. The use of the term ‘resident population is clearer and more accurate in our context. We regarded the population data as resident population, since the adopted Worldpop dataset was generated in a ‘top-down’ estimation whose most important source is population and housing censuses. We have added more explanations in subsection ‘2.3. Data Requirements’.
Also, in the methodology adopted accessibility is a key component that influences the "attraction" of the scale of analysis in question. This issue of accessibility should be explored in the assumptions and in the discussion.
Response:
Thank you for your suggestions. Indeed, accessibility is a crucial component influencing the "attractiveness" of analysis across different spatial scales. However, the utilization of accessibility in constructing LMU factors is merely one approach to factor construction and is not the primary focus of our study. Furthermore, the actual impact of factors related to accessibility on resident population distribution can be directly inferred through the analysis process by examining the q-statistic values of relevant factors. Specific analysis results regarding the influence of factors related to accessibility across different spatial scales can be found in the newly added subsection 3.3.4, as well as in related tables.
The authors use interchangeably the terms methodology and framework. These are not the same thing. Please choose where you are contributing (is it a methodology? Is it a framework?) and discuss how this contributes to the field. Be consistent with the use of the chosen term.
Response:
Thank you for the comments. We found them to be very helpful. We have refined the use of terminology to ensure greater focus and accuracy, and we have clarified the contribution of this research as the development of an explanatory framework.
Following your advice, we have expanded the descriptions in subsection 2.2 to provide greater clarity on the framework. From this perspective, the framework facilitates the selection of various representative factors tailored to the researcher's specific goals. In our use case, we have defined 14 factors to examine resident population distribution/change.
The findings reported in the discussion section are not novel or unexpected. That access to transportation nodes, life convenience, shopping centers etc positively correlates with population is a well known and researched link in both quantitative and qualitative or policy oriented studies. So I would like to see the authors reporting on what the methodology reported brought new to existing knowledge. A suggestion is to integrate the current discussion section with the reporting of the findings, and in the discussion explore of these findings mean in terms of policy guidelines of how best to optimize the LUM, accessibility, population distribution and change, etc.
Response:
Thank you for the helpful suggestions. We have thoroughly rewritten the 'Discussions' and 'Conclusions' sections to accurately convey the contribution of this study. Within the sections, we’ve engaged in the discussion on LMU and its impact on resident population distribution/change, contextualizing our findings within relevant studies in the field.
Additionally, we have clarified the research hypotheses in Section 1. These hypotheses pertain to the MAUP effect between LMU and resident population distribution/change, as well as the optimal spatial scale of LMU factors' influence.
I would also suggest taking a step back from the listing in the conclusions and critically reflect on what you find using your methodology. For example, that "some factors merely show a strong effect at certain scale" is a trivial finding. Of course the influence of parks and squares or of shopping malls will be more felt at a certain scale than at another. Different land-uses have different impacts, any urban planner would know this. But how can your methodology be used to optimize how to allocate these functions - so think at the level of how a planner or a policy maker could utilize your approach to optimize land-use allocation decisions.
Response:
Thank you for the comments. We found this very important and helpful. The whole section of ‘Conclusion’ is rewritten. It elaborates our development of this open-structured explanatory framework that helps to work with a selected set of LMU-related factors to reveal the optimal scale to discuss it and the key factors and their interactions in pairs. It, therefore, could work in different contexts for various planning aims and support planners and policymakers to identify the spatial scale and factors for more efficient implementation approaches to realise the aims of resident population distribution and changes via LMU.
Reviewer 3 Report
Comments and Suggestions for Authors
1) The phrase "the trans-scaled effect of LMU on population issues" in the description of the main objective of the paper is not quite precise. What population issues are in question (population density? quality of life? social diversity? population changes?). Furthermore, how does the purpose of the paper formulated in this way relate to the other assumptions of the study, including the research hypotheses?
2) The description of the Framework is very short and needs to be expanded. It is not clear on what basis the selection of representative factors was made or what factors were also considered.
3) The description of Data types is also very laconic and is limited only to listing the source of the data, without going into its specifics, reliability, or limitation. Many doubts are raised especially about the population data, which is only known to be from the Wordpop database. This is well illustrated by the discrepancy between the data from this database and the census data, as mentioned by the authors (lines 277-280, Table 2). The limitations of the OpenStreetMap database, which is created by the users, are also not discussed.
4) The Discussion section actually contains the most important conclusions of the study. There is no reference to the other studies addressing the impact of LMU on population distribution and changes. This section needs to be completely rewritten to reinforce the significance of the results obtained in the study and put them in a broader context.
5) The use of feature symbols in the text, rather than their full description, is debatable. This makes reading the text and interpreting the results much more difficult.
6) All the figures in the article are basically illegible. Figures 3 and 4 lack legends. It is not clear what values are expressed on the y-axis in Fig. 8 and Fig. 10.
Author Response
The main changes have been highlighted in the text.
1)The phrase "the trans-scaled effect of LMU on population issues" in the description of the main objective of the paper is not quite precise. What population issues are in question (population density? quality of life? social diversity? population changes?). Furthermore, how does the purpose of the paper formulated in this way relate to the other assumptions of the study, including the research hypotheses?
Response:
Thank you for bringing up this critical issue. The authors greatly appreciate your input. Our study is centered on the analysis of resident population distribution and its changes influenced by LMU factors, and assessing their impact across various spatial scales. We have diligently revised the manuscript to refine the objective, and the revised content has been highlighted. Specifically, we have replaced the term "trans-scaled effect" with "multi-scale effect," with its definition now included in subsection 2.1. Additionally, we have reorganized the research hypotheses in Section 1. These hypotheses pertain to the MAUP effect between LMU and resident population distribution/change, as well as the optimal spatial scale of LMU factors' influence.
Accordingly, we have adapted the title of this manuscript.
2)The description of the Framework is very short and needs to be expanded. It is not clear on what basis the selection of representative factors was made or what factors were also considered.
Response:
Thank you for your comments. Following your advice, we have expanded the descriptions in subsection 2.2 to provide greater clarity on the framework. From this perspective, the framework facilitates the selection of various representative factors tailored to the researcher's specific goals. In our study, we have defined 14 factors to examine resident population distribution, as detailed in subsections 2.1 and 2.3.
3)The description of Data types is also very laconic and is limited only to listing the source of the data, without going into its specifics, reliability, or limitation. Many doubts are raised especially about the population data, which is only known to be from the Wordpop database. This is well illustrated by the discrepancy between the data from this database and the census data, as mentioned by the authors (lines 277-280, Table 2). The limitations of the OpenStreetMap database, which is created by the users, are also not discussed.
Response:
Thank you for highlighting the limitations of the data. Indeed, the accuracy of data can introduce a certain degree of analytical error, which is inevitable when utilizing open datasets. However, our focus in this paper does not lie in addressing the inherent issues associated with the usage of such data. Instead, we analyze the trends presented by limited open data sources and their corresponding impacts, as evidenced by the growth trends illustrated in Table 2. Additionally, our framework is adaptable to incorporate the usage of other more precise datasets. Therefore, we believe that the issue of data accuracy is not the primary focus of our discussion.
Besides, we have provided additional explanations regarding the motivation and background of our data usage, as well as the top-down estimation approach of Worldpop population data, as detailed in Section 2.3.
4) The Discussion section actually contains the most important conclusions of the study. There is no reference to the other studies addressing the impact of LMU on population distribution and changes. This section needs to be completely rewritten to reinforce the significance of the results obtained in the study and put them in a broader context.
Response:
Thank you for highlighting this crucial issue. Following your advice, we've thoroughly rewritten the 'Discussions' section to enhance readability. Within this section, we’ve engaged in the discussion on LMU and its impact on resident population distribution, contextualizing our findings within relevant studies in the field.
5) The use of feature symbols in the text, rather than their full description, is debatable. This makes reading the text and interpreting the results much more difficult.
Response:
Thank you for your feedback. We understand that the abbreviations may pose challenges for readers. However, we aim to simplify text visualization in figures to avoid excessive verbosity, thus we find it necessary to retain them. Nonetheless, we have endeavored to consistently use the full names of variables throughout the manuscript to enhance readability.
6)All the figures in the article are basically illegible. Figures 3 and 4 lack legends. It is not clear what values are expressed on the y-axis in Fig. 8 and Fig. 10.
Response:
Thank you for your comments. We have replaced the ambiguous figures with higher-quality versions (Figures 3, 4, 5, 6, 7, 8, 10, and 12). The legends were added to Fig. 3 and 4. For Figures 8 and 10, we have included captions and explanations for the y-axis (i.e., the percentage of q values of factors relative to the benchmark). The relevant explanations can be found in subsections 3.3.1 and 3.3.2.
Round 2
Reviewer 1 Report
Comments and Suggestions for Authors
The text of the article has become much better. The necessary suggestions have been added. The corrections were made optimally by the authors. The results are easier to read, but the maps are still not clear enough.
Author Response
Thank you for your comments. We have enhanced the quality and clarity of all the images and have uploaded source files for reference. Additionally, we have incorporated necessary visual aids in Figures 8, 9, 10, and 11 to ensure a more intuitive representation of our research findings. We also uploaded another file to included all the maps with high resolution.
Reviewer 3 Report
Comments and Suggestions for Authors
Many thanks to the Authors for critically addressing the comments and suggestions in the review. The changes made improve the quality of the article. The issue that needs to be improved at the stage of preparing the final version of the manuscript is the quality of the figures.
Comments on the Quality of English LanguageMinor editing of English language required
Author Response
Thank you for your comments. We have enhanced the quality and clarity of all the images and have uploaded source files for reference. Additionally, we have incorporated necessary visual aids in Figures 8, 9, 10, and 11 to ensure a more intuitive representation of our research findings.
Regarding the requested minor editing of the English language, we have meticulously proofread the manuscript to ensure there are no language issues.